# Radiosynovectomy for the Treatment of Chronic Hemophilic Synovitis: An Old Technique, but Still Very Effective

**DOI:** 10.3390/jcm11247475

**Published:** 2022-12-16

**Authors:** Emerito Carlos Rodriguez-Merchan, Hortensia De la Corte-Rodriguez, Maria Teresa Alvarez-Roman, Primitivo Gomez-Cardero, Victor Jimenez-Yuste

**Affiliations:** 1Department of Orthopedic Surgery, La Paz University Hospital, 28046 Madrid, Spain; 2Department of Osteoarticular Surgery Research, Hospital La Paz Institute for Health Research—IdiPAZ (La Paz University Hospital—Autonomous University of Madrid), 28046 Madrid, Spain; 3Department of Physical and Rehabilitation Medicine, La Paz University Hospital, 28046 Madrid, Spain; 4Department of Hematology, La Paz University Hospital, 28046 Madrid, Spain

**Keywords:** hemophilia, synovitis, radiosynovectomy, technique, results, complications

## Abstract

A radiosynovectomy (RS) should be indicated when recurrent articular bleeds related to chronic hemophilia synovitis (CHS) exist, established by clinical examination, and confirmed by imaging techniques that cannot be constrained with hematological prophylaxis. RS can be performed at any point in life, mainly in adolescents (>13–14 years) and adults. Intraarticular injection (IAI) of a radioactive material in children might be arduous since we need child collaboration which might include general anesthesia. RS is our initial option for management of CHS. For the knee joint we prescribe Yttrium-90, while for the elbow and ankle we prescribe Rhenium-186 (1 to 3 IAIs every 6 months). The procedure is greatly cost efficient when compared to surgical synovectomy. Chemical synovectomy with rifampicin has been reported to be efficacious, inexpensive, simple, and especially practical in developing countries where radioactive materials are not easily available. Rifampicin seems to be more efficacious when it is utilized in small joints (elbows and ankles), than when utilized in bigger ones (knees). When RS and/or chemical synovectomy fail, arthroscopic synovectomy (or open synovectomy in some cases) should be indicated. For us, surgery must be performed after the failure of 3 RSs with 6-month interims. RS is an effective and minimally invasive intervention for treatment of repeated articular bleeds due to CHS. Although it has been published that the risk of cancer does not increase, and that the amount of radioactive material used in RS is insignificant, the issue of chromosomal and/or deoxyribonucleic acid (DNA) changes remains a concern and continued surveillance is critical. As child and adulthood prophylaxis becomes more global, RS might become obsolete in the long-term.

## 1. Introduction

Hemophilic arthropathy happens because of repeated bleeding into articulations resulting in swelling and degeneration of cartilaginous and osseous tissues in the affected joint. Even though hematological prophylaxis averts arthropathy, it is not always appropriate or accessible [1,2,3,4]. The only approach for averting arthropathy in people with hemophilia (PWH) without inhibitors is early primary prophylaxis, although it is not always entirely successful in avoiding articular problems [1,2,3,4]. In infants with inhibitors, prophylaxis with bypassing agents (aPCCs and/or RFVIIa) is also recommended to prevent articular complications [1,2,3,4]. To prevent joint degeneration in the hemophilic joints due to the impact of blood on the synovial membrane and the cartilage cells, early primary prophylaxis (intravenous infusion of the deficient factor) is the gold standard of treatment of hemophilia. In patients with hemophilia A (deficit of factor VIII), emicizumab prophylaxis has led to greater treatment satisfaction compared with FVIII prophylaxis, reflecting in part the low treatment burden of emicizumab associated with its infrequent, subcutaneous administration. Emicizumab can also be used in patients with inhibitors [5,6,7,8].

Articular bleeds cause chondrocyte death and chronic hemophilia synovitis (CHS) resulting in a malicious circle of synovitis-hemartrosis-synovitis. This circle has to be broken quickly to halt or slow the appearance of hemophilic arthropathy. The hypertrophied synovium can be detected through palpation as a hard mass. Removal of the hypertrophied synovial membrane may be performed by using radioisotopes [9,10,11,12,13,14,15,16,17,18,19,20,21]. Figure 1 summarizes the mechanism of action of radioactive materials injected intra-articularly (radiosynovectomy-RS) [21,22,23,24].

The objective of RS is to lower the danger of CHS (Figure 2) and recurrent hemarthroses that eventually degenerate the joint (hemophilic arthropathy). Hemophilia is a polyarticular disease, impacting mainly elbows, knees and ankles.

Therefore, it is important to remember that we are facing a multiarticular condition. Confirmation of the problem has to be done with magnetic resonance imaging (MRI) (Figure 3) and/or ultrasonography (US) (Figure 4).

The purpose of this article is to update the function of radiosynovectomy (RS) in the treatment of CHS in PWH.

## 2. When Should a Radiosynovectomy (RS) Be Indicated?

RS is the elimination of the hypertrophied synovium using an intraarticular injection of a radioisotope. We indicate a RS in the following circumstances [15,16,17,18,19,20]: (1) Two or more events of hemarthrosis in the preceding 6 months; hypertrophied synovium must be confirmed by MRI and/or US. (2) An additional RS must be performed in PWH with two or more episodes of articular bleed in the following 6 months. RS must only be done in specialized hemophilia centers.

MRI and/or US may enhance our prompt detection of CHS, and they can be performed at any stage in life. According to Doria et al. [25], even though MRI is the gold standard, US is highly helpful for assessing CHS. MRI can be performed once or twice a year, while US can be carried out as many times as needed. Sierra-Aisa et al. compared US and MRI in PWH [26]. It was found that US was valuable in uncovering joint bleeds, CHS and articular erosions, with results comparable to those of MRI.

When RS has to be repeated, the procedure is identical to that performed for the first procedure. The result measures have to be obtained 6 months after each RS and then every 6 months until the last follow-up evaluation. The most important result measures are the amount of hemarthroses per month (reduction in hemarthroses), factor use, and the clinical outcome [range of motion (ROM) of the involved joint].

Chemical synovectomy with rifampicin has been reported to be efficacious, inexpensive, simple, and especially practical in developing countries where radioactive materials are not easily available [27,28,29,30]. Rifampicin seems to be more efficacious when it is utilized in small joints (elbows and ankles), than when utilized in bigger ones (knees) [27,28]. When RS and/or chemical synovectomy fail, arthroscopic synovectomy (or open synovectomy in some cases) should be indicated. In a study, 6.3% of articulations required arthroscopic synovectomy or total knee arthroplasty (TKA) [20].

## 3. RS in Individuals with Inhibitors

Patients with inhibitors experience more bleeding episodes. As a consequence, they suffer greater ROM (range of movement) limitation, more movement impairments, more serious orthopedic complications, and poorer quality of life (QoL) [31,32].

Prophylaxis with bypassing drugs has proved its effectiveness in numerous reports. Up to now, aPCCs (activated prothrombin complex concentrates) and rFVIIa (recombinant factor VII activated) have been utilized in many patients. Both bypassing drugs have shown a decrease in the frequency of bleeding and amelioration of QoL [33,34,35,36,37,38,39,40,41,42,43,44,45,46,47,48,49,50,51,52,53]. In patients with hemophilia A, prophylaxis with subcutaneous emicizumab has also proved its efficacy in many publications [5,6,7,8].

When in patients with inhibitors, it is impossible to control recurrent bleeding using bypassing drugs or emicizumab as prophylaxis, a RS must be indicated. A study analyzed four individuals (6 RSs) aged 13 to 17 years who had 7 to 14 bleeding episodes per patient in the previous 12 months) [54]. No intraarticular bleeding or local inflammatory reaction was noticed during or after treatment, and no radioactivity was detected in the urine. All patients improved both subjectively and objectively. At a 2 year follow-up, the amount of bleeding events per year ranged from 1 to 5, a striking reduction.

In five PWH with inhibitors younger than 15 years, 13 articulations were treated with intraarticular injections of radioactive gold by Lofqvist and Petersson [55]. Of the 13 articulations injected, a bleeding-free interim of more than 6 months was obtained in 9 patients, of which 6 were free from bleeding for more than a year.

In a study, nine PWH with factor inhibitor aged from 3 to 4 years, 19 joints were treated with RS using radioactive gold [56]. RS was performed when the antibody titer was low (<10 Bethesda units). At long-term follow-up (range, 18–182 months), results were good in five joints, fair in one joint, and poor in eleven joints. The results were inferior to those for PWH without inhibitor.

We have previously stated that prophylaxis is essential in attempting to avoid the appearance of CHS, and that preeminent treatment for CHS in PWH with inhibitors is RS. With both strategies (prophylaxis and RS), the appearance of serious joint degeneration can be delayed [57].

According to Pasta et al., in PWH with inhibitors, a more serious degree of CHS is often observed because treatment is more difficult in this context [58]. For them, the best management option for recurrent hemarthroses and/or CHS is both chemical synovectomy and RS, with a success percentage of about 80% for both. Nevertheless, RS should be chosen in PWH with inhibitors because it makes it feasible to attain excellent fibrosis of the synovium commonly with one injection; without the necessity of more injections, the risk of recurrent hemarthroses and concentrate use diminishes.

Table 1 summarizes the most important articles on the subject [59,60,61,62,63,64,65,66,67,68,69,70,71,72,73,74,75].

## 4. Technique of Radiosynovectomy

RS must be performed under factor coverage to avoid the risk of hemarthrosis during the procedure. The main radioisotope used in the literature are 90Y (Yttrium-90), 186Rh (Rhenium-186) and 32P (Phosporus-32). All of them give off beta radiation and their therapeutic penetration powers (TPP) in millimeters are 2.8 mm, 1 mm, and 2.2 mm, respectively. In the knee we use Yttrium-90 [185 Megabecquerels (MBq)]. Rhenium-186 is used for elbows (56–74 MBq) and ankles (74 MBq). A small quantity of 99Tc (technetium) is introduced to perform articular scintigraphy after the procedure (to confirm the right dispersion of the radioisotope into the joint) [15,16,17,18,19,20].

We do not use local anesthetic. An ordinary needle is sufficient. When the joint has been accessed, all the fluid (blood or synovial fluid) is evacuated, and only then the radioisotope is introduced. The needle must be removed gradually whilst simultaneously introducing an anti-inflammatory agent (e.g., betamethasone), so as to not cause skin burn. Figure 5 shows the technique and all the elements required for ankle RS.

## 5. Effectiveness of Radiosynovectomy

In relevant articles, 40% to 85% of articulations achieve good clinical outcomes; between 30% and 80% of joints exhibit a reduction in hemarthroses; and 35% to 85% of individuals exhibit a reduction in factor use; in other words, the amount of joint bleeds per month decrease from 3 to 6 on average before the procedure to 1 after the procedure [9,10,11,12,13,14,15,16,17,18,19,20,21].

Between 1976 and 2013, we performed 500 RSs (with Yttrium-90 or Rhenium-186) in 443 joints of 345 individuals suffering from CHS [20]. Their average age was 24 years (range, 6–53) and the follow-up was 18.5 years on average (range: 6 months-38 years). We performed 1 to 3 procedures with a 6-month interim between them. Articular bleeding frequency declined by 64% on average.

In another report, we encountered that decrease of the number of hemarthoses after RS was 68% on average when RS-1 was used, 62% with RS-2, and 61% with RS-3 [16]. The volume of the synovium declined 31%. World Federation of Hemophilia (WFH) clinical score improved 19%. WFH radiological score did not improve [14,16]. In one of our reports, we found that knees required more RSs than elbows or ankles, and that the more serious hypertrophied synovial membranes required a greater number of RSs [15].

In another report, we found that RS was effective in all cohorts of patients, separately from the existence of inhibitor, the kind of articulation affected, the grade of CHS, and the existence of articular destruction (arthropathy) in the radiological examination [17]. In our center, we have also demonstrated that each RS performs separately in CHS [18]. In another report, we found that the variables analyzed improved to an equal grade in articulations with joint destruction in simple radiography (AJDSR) and without AJDSR. No joint without AJDSR required RS-3; this was the only dissimilarity our investigation found between joints without AJDSR and those with AJDSR when RS was performed [19].

In 2001, Silva et al. reported 130 RSs utilizing Phosporus-32 with a mean follow-up of 36 months [9]. Excellent and good outcomes (hemarthrosis decrease from 75% to 100%) were obtained in 79.2% of patients at 6 months to 8 years. No correlation between results and age or degree of arthropathy was found. No complications were observed.

In another report of 125 RSs, 54% got complete arrest of hemarthroses. 73% of patients reported improved mobility of the injected articulation. 79% of patients had a substantial amelioration in QoL attributable to the treated joint. No complications were observed [10].

The results of RS with 90Y in 163 joints of PWH were published by Heim et al. [11]. The median age at the time of the injection was between 11 and 15 years and the median follow-up period was 11 years. Over 80% of PWH reported a decrease in the amount of articular bleeds and 15% experienced full cessation of hemarthroses.

Mortazavi et al. analyzed 66 Phosporus-32 RS in 53 patients [12]. The mean follow-up was 31 months. The mean age of patients at the time of RS was 16 years. It was found that 77% of individuals experienced a 50% decrease in bleeding incidence after RS.

In 2009, Calegaro et al. evaluated the effectiveness of RS with 153-Sm-HA (185 Mbq) in 31 patients (30 males). Their mean age was 20 years (8 to 34 years). The use of 153 Sm-HA in the treatment of CHS was effective for elbows and ankles, but less effective for knees [13].

In 2010, Cho et al. analyzed clinical outcomes and radiologic evaluation of 58 RS (53 patients) utilizing Holmium-166-chitosan complex in PWH [14]. The mean age of patients was 14 years. The mean follow-up was 33 months. After the injection, the mean frequency of bleeding of the elbow diminished from 3.76 to 0.47 times a month, the knee from 5.87 to 1.12 times a month, and the ankle from 3.62 to 0.73 times per month, respectively.

Turkmen et al. performed 67 Yttrium-90 RS in 67 patients [21]. Their mean age was 17 years. The mean follow-up was 40 months. It was concluded that Yttrium-90 RS in the knee joint is an important resource for the treatment of CHS, markedly reducing joint bleeding.

## 6. Complications of Radiosynovectomy

Our reported percentage of adverse events is 1%. The adverse events that we have encountered are the following [15,16,17,18,19,20]: (1) Little skin burns repaired in 1–2 weeks just by cleaning them. They occur when the radioactive material is unintentionally introduced out of the joint (Figure 6); (2) infection (septic arthritis) which requires surgical management (arthrotomy and joint debridement) plus intravenous antibiotics; (3) swelling following injections solved with rest and NSAIDS (Nonsteroidal Anti-inflammatory Drugs). We specifically recommend cyclooxigenase-2 (COX-2) inhibitor inhibitors [76,77].

## 7. Is Radiosynovectomy Safe?

A great number of hemophilic children who may profit from RS for the constraint of CHS do not experience the procedure because there is debate in the literature concerning the safety of radioactive materials after two cases of acute lymphocytic leukemia (ALL) in infants with hemophilia managed with Phosphorus-32 RS were published [7].

In 2007, Turkmen et al. studied the genotoxic impact on the peripheral blood lymphocytes possibly caused by Yttrium-90 in children who were experiencing RS for CHS, using chromosomal aberration analysis (CA) and the micronuclei (MN) assay for detecting chromosomal aberrations, as well as the sister chromatid exchanges (SCE) technique for assessed DNA damage [78]. The outcomes of this investigation indicated that high radiation doses were not attained by peripheral lymphocytes of children who experience Yttrium-90 RS.

No augmentation in the risk of cancer has been reported by Infante-Rivard et al. [79]. Moreover, they found no dose-response relationship with the amount of radioisotope administered or number of RSs. Table 2 shows that the amount of radiation used is insignificant.

Infants experiencing knee RS get a radiation dose of approximately 0.74 mSv (90 megabecquerels-MBq) and for elbow and ankle RSs a dose of approximately 0.32 mSv (30–40 MBq). The radiation dose from natural sources is approximately 2 mSv and the recommended limit for patients (apart from natural sources) is 1 mSv/year. The lifetime cancer risk increases about 0.5% per 100 mSv/year.

## 8. Conclusions

The recommendation for a RS is the existence of recurrent hemarthroses due to CHS (verified clinically and by imaging techniques) that cannot be constrained with hematological prophylaxis. RS can be performed at any age, ideally in teenagers (>13–14 years). We advise Yttrium-90 for the knees and Rhenium-186 for elbows and ankles (1 to 3 RSs with 6-month interim). Chemical synovectomy with rifampicin has been reported to be efficacious, inexpensive, simple, and especially practical in developing countries where radioactive materials are not easily available. Rifampicin seems to be more efficacious when it is utilized in small joints (elbows and ankles), than when utilized in bigger ones (knees). When RS and/or chemical synovectomy fail, arthroscopic synovectomy (or open synovectomy in some cases) should be indicated. For us, surgery must be advised when three RSs with 6-month intervals fail to control CHS. RS is an effective and minimally invasive intervention for treatment of repeated articular bleeds due to CHS.

## Figures and Tables

**Figure 1 jcm-11-07475-f001:**
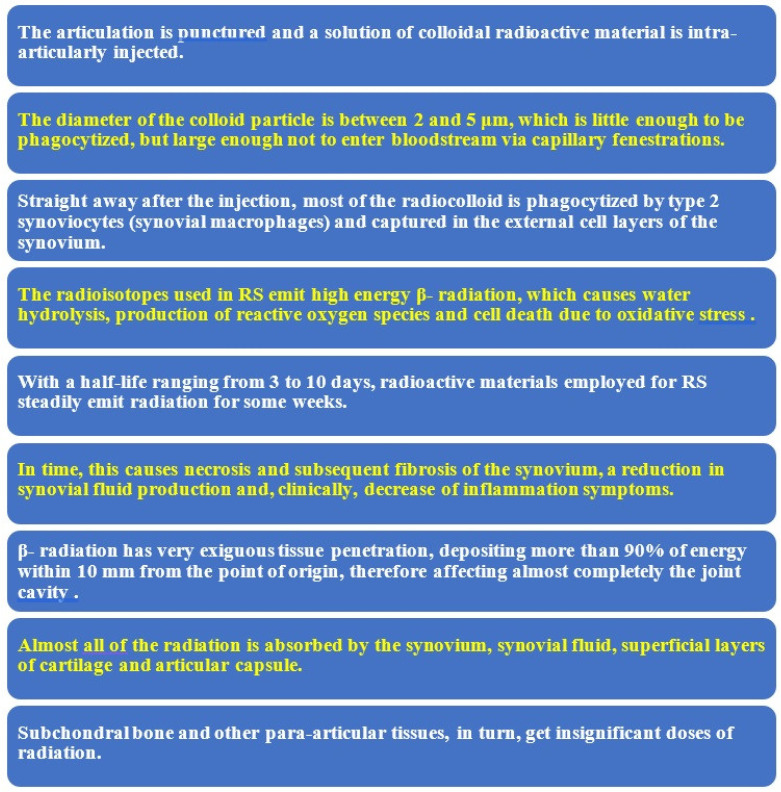
Mechanism of action of radiosynovectomy (RS).

**Figure 2 jcm-11-07475-f002:**
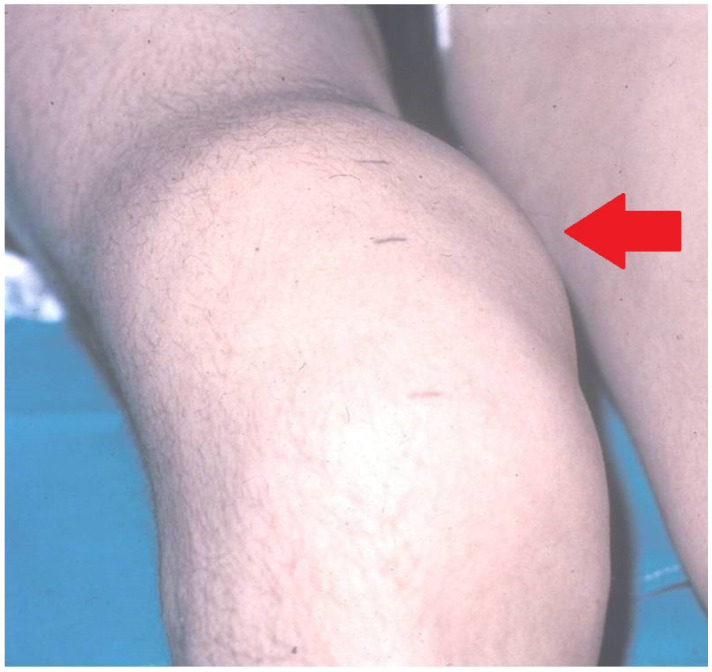
Serious chronic hemophilia synovitis (CHS) in a hemophilic patient (arrow).

**Figure 3 jcm-11-07475-f003:**
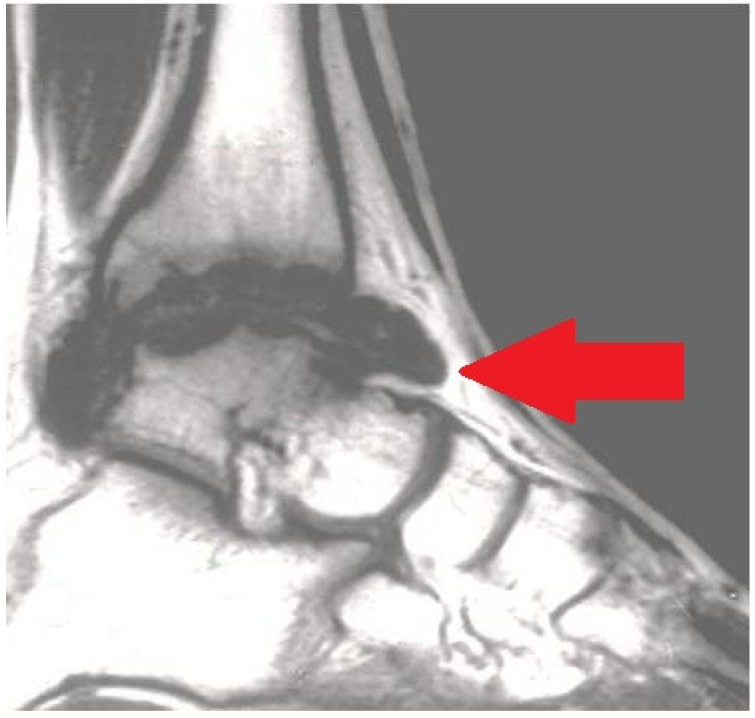
Magnetic resonance imaging (MRI) exhibiting serious chronic hemophilic synovitis (CHS) of the ankle (arrow).

**Figure 4 jcm-11-07475-f004:**
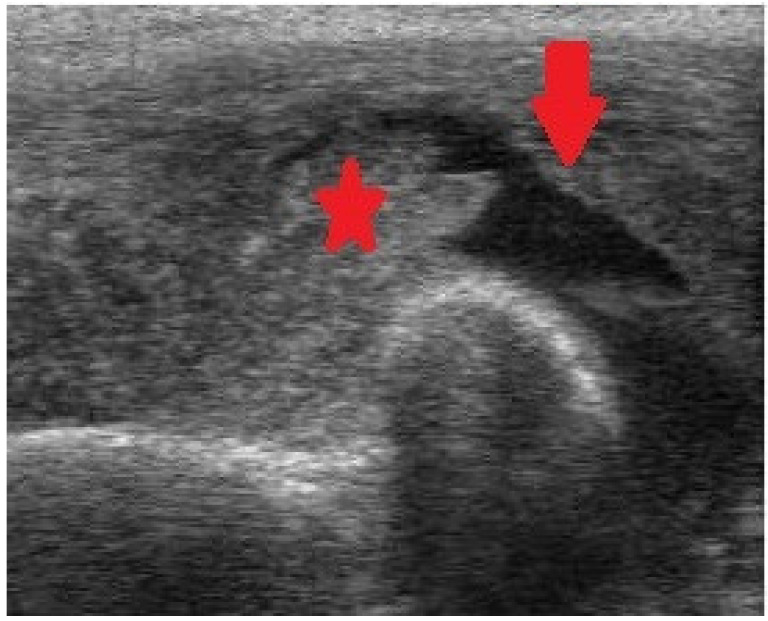
Ultrasonography (US) of the elbow exhibiting intraarticular fluid (arrow) and intense synovial enlargement (asterisk).

**Figure 5 jcm-11-07475-f005:**
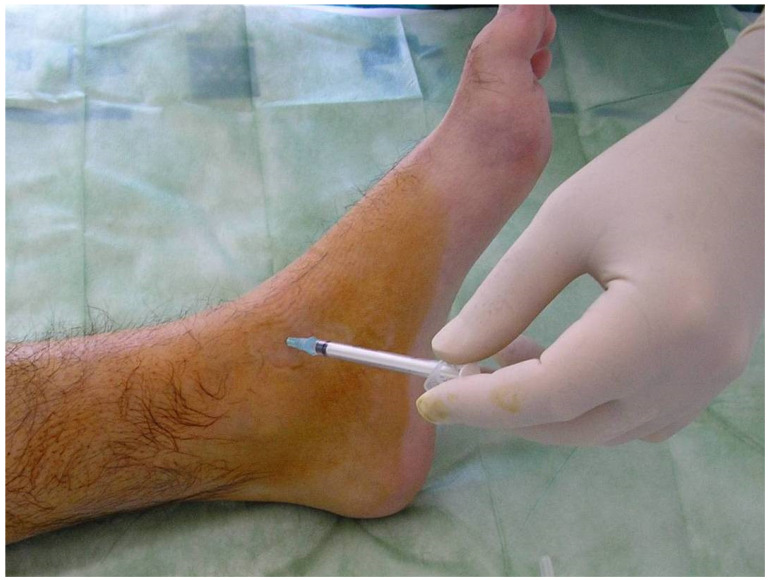
Radiosynovectomy (RS) of the ankle joint with Rhenium-186 in a haemophilic patients. The needle has to be removed gradually while at the same time introducing an anti-inflammatory agent.

**Figure 6 jcm-11-07475-f006:**
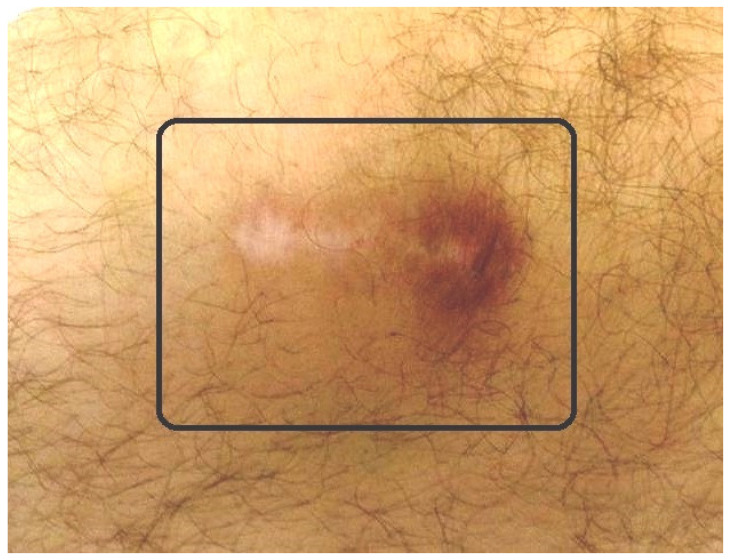
Small radioactive burn of the adjacent skin (square) after knee radiosynovectomy (RS).

**Table 1 jcm-11-07475-t001:** Reports on radiosynovectomy (RS) in people with hemophilia (PWH) from 2014 to 2022.

Authors [Reference]	Year	Results	Conclusions
Ozcan [59]	2014	This review focused on the practical aspects of RS in PWH.	RS rendered elimination of inflamed synovial membrane.
Rodriguez-Merchan [60]	2014	RS was the best approach for PWH with unremitting CHS of the knee irresponsive to a 3-month trial of hematological prophylaxis	No neoplastic changes were encountered.
Turkmen et al. [61]	2014	These authors reported their 10-year experience (2002–2012) of Yttrrium-90 RS in 82 knee joints of PWH (N = 67) with hemophilic synovitis. The mean age was 17 years, and the mean follow-up period was 40 months.	Y-90 RS in knee joint markedly diminished joint bleeding and long-run durability.
Rodriguez-Merchan and Valentino [62]	2015	This review analyzed the safety of RS in children with hemophilia and rendered a risk-benefit evaluation. Children undergoing knee RS receive a radiation dose of around 0.74 mSv (90 megabecquerels-MBq) and elbow and ankle RSs a dose of about 0.32 mSv (30–40 MBq).	RS must be indicated in children with inhibitors or in patients without inhibitors when bleeding is recurrent despite adequate factor replacement.
Rodriguez-Merchan et al. [63]	2016	Seventy RSs were carried out in 70 articulations (44 elbows, 26 ankles) of 70 PWH diagnosed with chronic synovitis. The mean patient age was 20 years. RS led to substantial improvement in the three variables analyzed (six months prior to RS vs. six months after RS), namely in the number of episodes of hemarthrosis (67.8% amelioration); the size of the synovial membrane as measured by means of a clinical scale (43.8% improvement) and imaging techniques in millimeters (26.7% amelioration).	Yttrium-90 RS and Rhenium-186 RS were equally efficacious in diminishing the number of hemarthroses and the size of the synovial membrane in ankles and elbows in the short-run (6 months).
Zhang et al. [64]	2016	In 24 knees assessed in PWH (N = 22), there was a substantial decrease in the number of hemarthroses after Phosporus-32 colloid treatment, along with substantial pain alleviation.	The frequency of hemarthroses was substantially diminished in the short-run by RS
Wang et al. [65]	2017	RS with Phosporus-was carried out in PWH (N = 326, 405 joints). Synovial volumes diminished after 6 months when compared with baseline.	RS was a safe and efficacious technique.
McGuinn et al. [66]	2017	In the ATHNdataset these authors found 19 539 control-patients and 196 case-patients treated with RS.	Case-patients had worse joint ROM compared to control-patients.
Sabet et al. [67]	2017	PWH (N = 34) experienced RS after failure of conservative treatment in 34 joints (8 knees, 5 elbows, 21 ankles). Joint bleeding frequency (hemarthrosis) diminished from 4.5 to 2.1 during the first 6 months after RS. No significant amelioration was found for ROM.	Hemophilic synovitis can be effectively treated with RS.
Gallant et al. [68]	2018	These authors described a boy with severe hemophilia A, who suffered arterial vasculitis and perivasculitis targeting the brachiocephalic, right common carotid, and right subclabvian arteries happening within few days after Phosporus-32 RS.	This complication was possibly due to RS.
Rodriguez-Merchan [69]	2019	This review article stated that RS was a simple, efficacious and safe technique for the restrain of CHS that produces recurrent hemarthrosis. RS should be the first invasive option (instead of arthroscopic synovectomy) for treatment of CHS.	RS must be carried out under factor coverage as soon as possible.
Oliveira et al. [70]	2019	This study analyzed 119 family members’ safety (16.7% pregnant women). Results demonstrated that family members should be recommended to stay at 1 m from PWH to diminish accumulated dose by 98%.	RS was a safe procedure for family members.
Kachooei et al. [71]	2020	PWH (N = 20) were assessed before RS, and at 1, 3, 6 and 12 months after RS with Rhenium-188. Minor adverse events, including temporary pain and swelling happened in 20% of PWH, and no serious adverse events were found after Rhenium-188 RS.	Rhenium-188 was a good treatment for PWH with recurrent hemarthrosis.
Koc et al. [72]	2020	RS was carried out in 51 articulations of 22 PWH with inhibitors diagnosed with CHS. The mean bleeding frequency of the joints was 11 within the last 6 months in the pre-injection assessment. After the injection, the mean bleeding frequency of the joints diminished to 1 for first 6 months.	RS was an effective and safe technique in PWH with inhibitors.
Ebrahimpour et al. [73]	2020	These authors presented the long-term follow-up of 32 Phosporus-32 RS performed in 44 patients (52 RSs). The mean follow-up was 15 years. The joint bleeding frequency was not statistically significant at the latest follow-compared with 31 months (0.8 vs. 0.4 per week).	The bleeding control effect of Phosporus-32 RS on the target joint remained over time.
Magalhaes et al. [74]	2021	In a one year follow-up, 22 PWH (25 joints) who presented 3 hemarthroses or more in the same joint over the last 6 months experiencedYttrium-90 RS.	The volume of the synovium was diminished after the RS.
Szerb et al. [75]	2021	This study analyzed the role of RS in the restraint of hemarthroses in PWH (N = 54). Mean age of the patients was 32 years. The RS led to a 95% decrease in hemarthroses per year and eliminated the incidence of further hemarthroses in 55% of the treated articulations.	The findings of this study supported the view that RS can be considered as the first choice treatment for posttraumatic joint bleedings of PWH.

ROM = Range of motion; CHS = Chronic hemophilic synovitis.

**Table 2 jcm-11-07475-t002:** Estimation of the dose of radiation of RS in various situations. Notice that the radiation dose of RS is insignificant.

Natural Sources	2 mSv
Advised limit for patients (apart from natural sources)	1 mSv per year
Chest X-ray	0.1 mSv
Body CT scan	10 mSv
Bone scintigraphy (Tc-99) in individual of 70 kg	5.6 mSv (700 MBq)
Increased lifetime cancer risk about 0.5%	100 mSv per year
The individual could suffer from radiation illness. It is often deadly and can cause bleeding, shedding of the lining on the gastrointestinal tract, and increase cancer risk (DNA mutation)	2000 mSv
Knee RS	Adulthood: 1.48 mSv (185 MBq)
	Childhood: 0.74 mSv (90 MBq)
Elbow–ankle RSs	Adulthood: 0.54 mSv (56–74 MBq)
	Childhood: 0.32 mSv (30–40 MBq)

RS = Radiosynovectomy; mSv = millisieverts; CT = Computed tomography; Tc = Technetium; MBq = Megabecquerels; DNA = deoxyribonucleic acid.

## Data Availability

Not applicable.

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
