# Peer review of "Radiosynovectomy for the Treatment of Chronic Hemophilic Synovitis: An Old Technique, but Still Very Effective"

_jcm, 2022, doi:10.3390/jcm11247475_

Round 1
Reviewer 1 Report
Rodrigues-Merchan et al. provide an interesting review of radiosynovectomy for the treatment of chronic hemophilic synovitis. I agree that this old technique can still be useful.
Radiosynovectomy is currently the preferred procedure when radioactive materials are available, however, rifampicin is an effective alternative method if radioactive materials are not available. I suggest chemical synovectomy be a little more detailed: whether it may constitute an alternative, a second line therapy before arthroscopic synovectomy, and why, if not.
Currently, it is not possible to talk about prophylaxis in patients with inhibitors just referring to by-passing drugs without mentioning emicizumab.
Author Response
REVIEWER-1
Below you can see our answers to your important comments.
In the revised paper (REVISION-1) you can see IN RED the changes we have made to respond to your comments.
Rodrigues-Merchan et al. provide an interesting review of radiosynovectomy for the treatment of chronic hemophilic synovitis. I agree that this old technique can still be useful.
Radiosynovectomy is currently the preferred procedure when radioactive materials are available, however, rifampicin is an effective alternative method if radioactive materials are not available. I suggest chemical synovectomy be a little more detailed: whether it may constitute an alternative, a second line therapy before arthroscopic synovectomy, and why, if not.
AUTHORS: We have included some text (and References) on the role of rifampicin synovectomy.
Currently, it is not possible to talk about prophylaxis in patients with inhibitors just referring to by-passing drugs without mentioning emicizumab.
AUTHORS: We have included some text (and References) on the role of emicizumab prophylaxis.
Reviewer 2 Report
This is a review of a radiosynovectomy for the treatment of chronic hemophilic synovitis. The author did an extensive literature review and did a well summarize of evidences .
Author Response
REVIEWER-2
This is a review of a radiosynovectomy for the treatment of chronic hemophilic synovitis. The author did an extensive literature review and did a well summarize of evidences.
AUTHORS: We do really appreciate the positive evaluation of the Reviewer. Therefore, no changes gave been made in the text.
Reviewer 3 Report
The authors present a comprehensive review for the use of radiosynovectomy for hemophilia. The authors accurately present a literature review with patient outcomes following radiosynovectomy. This paper also comments on the concern for the development of cancer following radiosynovectomy. It is an interesting review and relevant for the care of patients with hemophilia.
Author Response
REVIEWER-3
The authors present a comprehensive review for the use of radiosynovectomy for hemophilia. The authors accurately present a literature review with patient outcomes following radiosynovectomy. This paper also comments on the concern for the development of cancer following radiosynovectomy. It is an interesting review and relevant for the care of patients with hemophilia.
AUTHORS: We do really appreciate the positive assessment of the Reviewer. Therefore, no changes have been made.